# Oxidized Alginate Dopamine Conjugate: In Vitro Characterization for Nose-to-Brain Delivery Application

**DOI:** 10.3390/ma14133495

**Published:** 2021-06-23

**Authors:** Adriana Trapani, Filomena Corbo, Gennaro Agrimi, Nicoletta Ditaranto, Nicola Cioffi, Filippo Perna, Andrea Quivelli, Erika Stefàno, Paola Lunetti, Antonella Muscella, Santo Marsigliante, Antonio Cricenti, Marco Luce, Cristina Mormile, Antonino Cataldo, Stefano Bellucci

**Affiliations:** 1Department of Pharmacy-Drug Sciences, University of Bari “Aldo Moro”, I-70125 Bari, Italy; filomena.corbo@uniba.it (F.C.); filippo.perna@uniba.it (F.P.); andrea.quivelli@uniba.it (A.Q.); 2Department of Biosciences, Biotechnologies and Biopharmaceutics, University of Bari “Aldo Moro”, I-70125 Bari, Italy; gennaro.agrimi@uniba.it; 3Dipartimento di Chimica and CSGI-Bari Unit, Università degli Studi di Bari Aldo Moro, I-70125 Bari, Italy; nicoletta.ditaranto@uniba.it (N.D.); nicola.cioffi@uniba.it (N.C.); 4Consorzio C.I.N.M.P.I.S., Via E. Orabona 4, I-70125 Bari, Italy; 5Dipartimento Scienze e Tecnologie Biologiche e Ambientali, University of Salento, I-73100 Lecce, Italy; erika.stefano@unisalento.it (E.S.); paola.lunetti@unisalento.it (P.L.); antonella.muscella@unisalento.it (A.M.); santo.marsigliante@unisalento.it (S.M.); 6ISM-CNR, Via del Fosso del Cavaliere 100, I-00133 Rome, Italy; antonio.cricenti@artov.ism.cnr.it (A.C.); marco.luce@artov.ism.cnr.it (M.L.); 7Istituto Nazionale di Fisica Nucleare-Laboratori Nazionali di Frascati, Via Enrico Fermi 54, Frascati, I-00044 Rome, Italy; c.mormile@gmail.com (C.M.); antonino.cataldo@lnf.infn.it (A.C.); 8Department of Chemical Science and Technologies, University of Rome Tor Vergata, Via della Ricerca Scientifica 1, I-00133 Rome, Italy

**Keywords:** dopamine, oxidized alginate, mucoadhesion, microscopy, cell viability

## Abstract

Background: The blood–brain barrier (BBB) bypass of dopamine (DA) is still a challenge for supplying it to the neurons of *Substantia Nigra* mainly affected by Parkinson disease. DA prodrugs have been studied to cross the BBB, overcoming the limitations of DA hydrophilicity. Therefore, the aim of this work is the synthesis and preliminary characterization of an oxidized alginate-dopamine (AlgOX-DA) conjugate conceived for DA nose-to-brain delivery. Methods: A Schiff base was designed to connect oxidized polymeric backbone to DA and both AlgOX and AlgOX-DA were characterized in terms of Raman, XPS, FT-IR, and ^1^H- NMR spectroscopies, as well as in vitro mucoadhesive and release tests. Results: Data demonstrated that AlgOX-DA was the most mucoadhesive material among the tested ones and it released the neurotransmitter in simulated nasal fluid and in low amounts in phosphate buffer saline. Results also demonstrated the capability of scanning near-field optical microscopy to study the structural and fluorescence properties of AlgOX, fluorescently labeled with fluorescein isothiocyanate microstructures. Interestingly, in SH-SY5Y neuroblastoma cell line up to 100 μg/mL, no toxic effect was derived from AlgOX and AlgOX-DA in 24 h. Conclusions: Overall, the in vitro performances of AlgOX and AlgOX-DA conjugates seem to encourage further ex vivo and in vivo studies in view of nose-to-brain administration.

## 1. Introduction

Parkinson’s disease (PD), the second most common neurological disorder (ND) after Alzheimer disease, is mainly characterized by the progressive loss of dopaminergic neurons in the *Substantia Nigra*, with a consequent dysregulation of motor activity [1]. From a clinical point of view, the death of dopaminergic neurons brings about classical PD symptoms, including bradykinesia, muscular rigidity, and tremors. On the other hand, the histopathological hallmark of the disease is the presence of Lewy bodies, i.e., abnormal aggregates of proteins of which α-synuclein is the main component [2].

One of the main problems to face in PD treatment, as well as in other NDs, is the transport of therapeutic agents across the blood–brain barrier (BBB), which, even though it is permeable, still limits the entrance of molecules into brain, which not only may cause neuronal damage, but also may have therapeutic usefulness [3]. Therefore, a very critical issue to treat NDs is to bypass the BBB.

In recent years, several research groups have highlighted that administration via the intranasal route of neurotherapeutics constitutes an advantageous approach to bypass the BBB, thus enabling direct access to the brain by olfactory and trigeminal neural connections between the nose and brain [4,5,6,7]. The most relevant advantage of using nose-to-brain drug administration is the minimally invasive feature of this pathway, leading to a high patient compliance; hence, the design of nasal formulations is crucial in order to address the active principles to the site of action [7]. For instance, the anatomic features of the olfactory cleft at the roof of the nasal cavity require that excipients with good mucoadhesion properties be good candidates to prolong the residence time at the mucosa so that drug bioavailability can be increased [7]. Among mucoadhesive polymers for nasal delivery, chitosan, carbopol, carboxymethylcellulose, polyacrylic acid, and others have been extensively described. In particular, semisolid formulations, such as hydrogels, have demonstrated their suitability in intranasal administration. Due to their molecular structure, hydrogels exhibit distinct advantages, being biocompatible, having low toxicity, and hydrophilic delivery systems. Moreover, their drug release kinetics can be tuned by controlling the crosslinking degree; some hydrogels can possess interesting features, including P-glycoprotein efflux pump (P-gp) inhibition and further biomedical applications [8,9,10]. In addition to hydrogels, nanoparticulate systems, such as polymeric nanoparticles, polymeric micelles and lipid-based formulations, have also been used in nose-to-brain delivery [7,11,12,13].

Currently, the main objective of PD therapy is to compensate for the loss of dopaminergic neurons and to restore suitable amounts of the neurotransmitter, dopamine (DA), which cannot cross the BBB. This approach is denoted as a “dopamine replacement strategy” and has been evaluated by loading different nanocarriers with both the free neurotransmitter (or its precursor L-dopa) and dopaminergic drugs [14,15,16,17,18]. A potential drawback of this strategy is the premature leakage of the therapeutic cargo from the nanocarrier, which can be minimized by linking the therapeutic agent to a polymeric backbone using a cleavable chemical bond. The resulting macromolecular conjugates possess several advantages, including a controlled release kinetic, prolonged circulation, and improved mucoadhesive properties [19,20,21]. In addition, they may provide free neurotransmitter- or dopaminergic drug-loaded nanoparticulate systems that are potentially able to cross the BBB via endocytosis [22,23].

The aim of the present work was to synthesize and characterize, from a physicochemical point of view, a novel macromolecular conjugate of DA, i.e., oxidized alginate (AlgOX) imine conjugate (AlgOX-DA), for potential intranasal DA delivery. Alginates (Alg), a class of natural-origin polysaccharides, are anionic linear block copolymers comprised of 1,4-linked -D-mannuronic acid (M) and -L-guluronic acid (G) (Figure 1), from which physical hydrogels are formed using solutions containing Ca^2+^ and Ba^2+^ divalent cations [24]. However, Alg-derived hydrogels are limited by their slow degradability in vivo, and oxidation of Alg leads to an increased biodegradability [24]. Moreover, a further advantage of AlgOX compared to Alg is that the former possesses additional organic functions that can be exploited in the conjugation process. For instance, in the oxidation process, two aldehyde groups are created, which can be useful in the conjugation reaction with a primary amino group containing substances [25]. Therefore, most interest has been focused on AlgOX for several biomedical applications. In this work, in addition to the synthesis of AlgOX-DA imine conjugate and its spectroscopic characterization, we describe the evaluation of its mucoadhesive properties, DA release kinetic in simulated nasal fluid and in phosphate buffer saline, as well as cytotoxicity by the human neuroblastoma SH-SY5Y cells line.

## 2. Materials and Methods

Sodium alginate (NaAlg, M/G = 1.49; Molecular weight = 396 kDa), sodium periodate (NaIO_4_), dopamine hydrochloride (DA), polyethylene glycol (Molecular weight = 200 g/mol), triethylamine (TEA), porcine stomach mucin (type II, bound sialic acid ~1%), and fluorescein 5(6)-isothiocyanate (FITC) were provided from Sigma-Aldrich (Milan, Italy). Hydroxyethyl cellulose (HEC, Natrosol 250) was purchased from Aakon Polichimica (Milan, Italy). According to the manufacturer instructions, a solution of HEC at a concentration of 2% in water provided a viscosity value equal to 5500 mPa·s. Dialysis tubes with a MWCO 3500 Da and 1200–14,000 Da were purchased from Spectra Labs (Rome, Italy). Throughout this work, double distilled water was used. All other chemicals used were of reagent grade.

### 2.1. Synthesis of Oxidized Alginate and Oxidized Alginate-DA

A total of 500 mg of NaAlg was dissolved in 5 mL of ethanol and then 530 mg of NaIO_4_, was added and dissolved in 5 mL of distilled water under stirring at r.t. and under dark conditions for 24 h. To quench the reaction, a few drops of polyethylene glycol were added, leaving the solution under stirring for a further 30 min.

The resulting product was purified by solubilization/precipitation according to the following procedure. Distilled water (5 mL) was added to the reaction mixture in order to solubilize the AlgOX, then NaCl (0.20 g) dissolved in ethanol (5 mL) was added to precipitate the NaAlg which was collected by filtration This solubilization/precipitation procedure was repeated two more times; the first by dissolving the filtrate AlgOX in distilled water (5 mL) and then adding NaCl (0.06 g) dissolved in ethanol (5 mL) to precipitate the AlgOX. Finally, dissolution of the filtrate in distilled water (5 mL) and addition of NaCl (0.03 g) and acetone (10 mL) occurred to precipitate AlgOX. Ethanol (5 mL) was poured to wash the collected final precipitate under stirring for 15 min and then the precipitate was dried at 45 °C under reduced pressure. A constant weight was obtained [26].
FT-IR (KBr) ν (cm^−1^): 3435 (OH), 2915 (HC=O), 1622 (C=O) 1388 (OH) cm^−1^.

The AlgOX-DA imine conjugate was prepared according to the synthetic procedure outlined in Scheme 1, following Gao et al.’s methodology [27]. A solution of AlgOX (25 mg) in distilled water (3 mL, pH 4) was poured into a solution of DA (68 mg) and TEA (55 μL) in distilled water (1 mL), in a light protected vessel under a nitrogen atmosphere. Then, the mixture underwent magnetic stirring in the dark for 6 h. The obtained dispersion was dialyzed in dialysis tubes (MWCO = 12,000–14,000 Da) and lyophilized for 96 h (Lio5Pascal, Italy) to obtain a brown powder as product in a yield of 62.8%.
AlgOX-DA: FT-IR (KBr) ν (cm^−1^): 3439 (OH), 2917 (HC=O), 2073 (aliphatic CH), 1636 (C=N) cm^−1^.

### 2.2. Quantitative Analysis of DA

The quantification of DA was performed using HPLC, as previously reported [28]. Precisely, the mobile phase was a mixture of 0.02 M potassium phosphate buffer, pH 2.8: CH_3_OH 70:30 (*v*/*v*), and the isocratic elution of the column occurred at a flow rate of 0.7 mL/min. In the HPLC chromatograms, the retention times of DA and AlgOX-DA were equal to 5.5 and 4.9 min, respectively.

### 2.3. Spectroscopic Characterization

#### 2.3.1. UV-Vis

For pure DA, AlgOX and AlgOX-DA, UV-vis spectra were acquired with a PerkinElmer Lambda Bio 20 spectrophotometer at a wavelength range of 230–340 nm. Prior to acquiring the spectra, pure DA aqueous solution was prepared at 0.05 mg/mL at 25 °C, and AlgOX and AlgOX-DA were dissolved, obtaining the final concentrations of 1 mg/mL and 0.25 mg/mL, respectively. A slight sonication was required for complete dissolution for AlgOX-DA.

#### 2.3.2. FT-IR Spectroscopy

FT-IR spectra were obtained with KBr discs using a Perkin Elmer 1600 FT-IR spectrometer (Perkin Elmer, Milan Italy). The analysis was carried out at room temperature (r.t.) in the range of 4000–400 cm^−1^ at a resolution of 1 cm^−1^ for the following samples: pure DA, AlgOX, and AlgOX-DA.

#### 2.3.3. Raman Spectroscopy

To record the Raman spectra an InVia microscope (Renishaw, Wotton-under-Edge, UK) was employed with a laser source set at a wavelength of 532 nm in the presence of a 100× objective. The specimens were placed onto a microscope slide. The range examined was 200–4200 cm^−1^ using a 1800 (L)/mm grating.

#### 2.3.4. Scanning Electron Microscopy (SEM) and X-ray Energy Dispersive (EDX) Analysis

SEM and EDX analyses were carried out using a VegaII microscope (Tescan, Czech Republic), with a Quantax elemental detector (Bruker, Billerica, MA, USA). The specimens were placed onto a microscope stub and a sputter coater (Cressington, UK) was used to cover specimens with a thin gold layer. Then, the specimens were introduced into the SEM chamber and analyses were performed using a high voltage of 20 kV.

#### 2.3.5. X-ray Photoelectron Spectroscopy (XPS) Analysis

To determine XPS spectra, a Versa Probe II scanning XPS microprobe spectrometer (Physical Electronics GmbH, Feldkirchen near Munich, Germany), equipped with a monochromatized Al Kα source (X-ray spot = 200 μm), with a working power of 50.3 W, was used. In order to acquire wide scans and detailed spectra, the fixed analyzer transmission (FAT) mode, with a pass energy of 117.40 eV and 29.35 eV, respectively, was employed. Charge compensation (1.1 V 20.0 μA) was obtained using an electron gun. Data processing was performed using MultiPak software v. 9.9.0.8 (Physical Electronics GmbH, Feldkirchen near Munich, Germany). 

### 2.4. NMR Spectroscopy

A Bruker 500-Hz spectrometer allowed the recording of ^1^H-NMR spectra and parts per million (δ) was used to report chemical shifts. Spectra were processed using Mnova (Mestrelab Research).

To allow the determination of AlgOX-DA composition via ^1^H NMR spectrometry, dissolution in D_2_O was performed for this sample at a concentration of 6 mg/mL and heated at 40 °C for few minutes before the analysis.

### 2.5. Differential Scanning Calorimetry (DSC)

Prior to perform thermal analyses, melting point determination for AlgOX and AlgOX-DA occurred via the use of the capillary method (Stuart Scientific SMP3 electrothermal apparatus, Bibby Scientific, Milan, Italy). DSC thermograms were obtained for DA, AlgOX, and AlgOX-DA working with a Mettler Toledo DSC 822e STARe 202 System equipped with DSC MettlerSTARe Software (version II, Mettler-Toledo, Milan, Italy). About 5 mg of each sample was placed in an aluminum pan and hermetically sealed. The scanning rate was set to 5 °C/min under an indium atmosphere (99.9%) and adopted for DSC calibration following the procedure of MettlerSTARe Software. Each experiment was carried out in triplicate.

### 2.6. Determination of Substitution Degree (DS) of AlgOX-DA

For substitution degree (DS) determination, the AlgOX-DA conjugate was subjected to acid hydrolysis. Briefly, 2 mg of AlgOX-DA conjugate were weighed and dissolved in HCl 1 N (pH 1) under stirring and light protection at r.t. for 3 h. Afterwards, the yellow liquid was collected, which was then centrifuged (16,000× *g*, for 45 min, Eppendorf 5415D, Germany) prior to performing HPLC analyses (Section 2.2) of the obtained supernatant. DS was calculated as mg DA/mg AlgOX-DA imine conjugate.

### 2.7. Mucoadhesion Tests

The evaluation of the mucoadhesive properties of the AlgOX and AlgOX-DA conjugate in simulated nasal fluid (SNF) was based on an in vitro method which used turbidimetric measurements, as described previously [21,28]. SNF composition included CaCl_2_·2H_2_O (0.32 mg/mL), KCl (1.29 mg/mL), and NaCl (7.45 mg/mL) at pH 6.0 [29].

Freshly prepared mucin dispersions in SNF (0.5 mg/mL) were incubated at 37 °C under stirring (150× *g*) for 24 h before starting the experiments. Separately, all samples were dispersed in SNF, providing a final concentration of 0.04% (*w*/*v*). To 6 mL of mucin dispersion in SNF, 6 mL of sample dispersion was added and the turbidity of the corresponding mixtures, incubated at 37 °C and under stirring (150× *g*), was determined at a wavelength of 650 nm using a Perkin-Elmer Lambda Bio 20 spectrophotometer at different time points (i.e., 0, 2, 5, 7 and 24 h). HEC (0.4 mg/mL in SNF) was employed as positive control. All experiments were carried out in triplicate.

### 2.8. In Vitro Release Studies

A total of 10–12 mg of AlgOX-DA (corresponding to 0.3–0.5 mg of DA) were collected in a dialysis bag (MWCO 3500 Da) containing 1.5 mL of double distilled water and then the dialysis bag was dispersed in 20 mL of SNF (or PBS) at 37 ± 0.1 °C in an agitated (40× *g*/min) water bath (Julabo, Milan, Italy). At scheduled time-points, 0.8 mL was withdrawn from the receiving medium and 0.8 mL of fresh medium was used to maintain sink conditions. Each sample underwent centrifugation (16,000× *g*, 45 min, Eppendorf 5415D, Germany), and the concentrations of DA were quantified in the resulting supernatant by HPLC, as reported in Section 2.2. For the cumulative release calculation, an approach reported in the literature was followed [30]. In details, the calculation of the accumulated release was based on the following equation:Q_n_ = C_n_ × V_0_+ (C_1_ + C_2_ + C_3_ + …. C _n−1_) × V(1)

Q_n_ is the cumulative release of the *n*-th sampling point, C represents the solution concentration in the SNF (or PBS), C_n_ is the concentration in the SNF (or PBS) of the *n*-th sampling point, V_0_ is the total volume of release medium, and V is the volume of each sample withdrawn.

All release experiments were carried out in triplicate.

### 2.9. Preparation of Fluorescent AlgOX

Following a previous protocol [31], after dissolving 30 mg of AlgOX in 3 mL of double distilled water, the pH value was adjusted to 5 by dropping in HCl 0.01N. Then, 1 mL of an ethanolic solution of FITC (5 mg/mL) was added and stirring was applied for 24 h under dark conditions at r.t. The mixture was then dialyzed in water using dialysis tubing (SpectraPore^®^Dialysis MWCO 12,000–14,000 Da) for 3 days and then freeze dried for 72 h (Lio Pascal 5P, Italy).

To determine the labeling efficiency of the resulting FITC-AlgOX conjugate, the fluorescence intensity of FITC-AlgOX dissolved in 0.1 M phosphate buffer, pH 8.0, was determined at a concentration of 2 μg/mL. The labeling efficiency (%) was the percent weight of FITC to weight of the FITC-polymer. The fluorometer was calibrated with standard solutions of 1 to 140 ng/mL of FITC in phosphate buffer, pH 8.0, arising from dilution in such a buffer of a stock solution of 100 μg/mL FITC in methanol (excitation and emission wavelengths of 488 and 525 nm, respectively; slits, 2.5 cm).

### 2.10. Photon Correlation Spectroscopy and SNOM Visualization

Particle size and polydispersity index for AlgOX and AlgOX-DA were determined \ using a Zetasizer Nano ZS (ZEN 3600, Malvern, UK) apparatus using the photon correlation spectroscopy (PCS) mode. Particle size and PDI were measured at concentrations of 1 mg/mL and 0.25 mg/mL for aqueous AlgOX and AlgOX-DA, respectively, introducing 1 mL of each sample into the cuvette.

For SNOM visualization, the apparatus setup employed consisted of a hybrid inverted/SNOM, a home-built SNOM connected to a conventional inverted optical microscope [32]. The instrument allowed the collection of topographic images simultaneously with optical data; both transmittance and fluorescence were recorded and the fluorophore signal had a typical absorbing peak above 400 nm and a wide-band emission above 500 nm. The 405 nm laser line was coupled in the Al-covered silica optical fiber and the fluorescence and transmitted light were collected by the 20× lenses of an Olympus microscope through a high-pass filter. The fluorescence signal was collected above 500 nm. This system worked with the inverted microscope to determine the features of the solutions of FITC-AlgOX (1 mg/mL and 0.01 mg/mL in water), to perform a far-field characterization and to pin-point the location where the high resolution SNOM measurements should be carried out. For analysis, FITC-AlgOX samples were deposited onto a glass substrate by means of a micropipette, dried, and examined using shear-force and fluorescence SNOM nanospectroscopy. Data were collected from several sample areas using different sets of experiments, and then analyzed using Gwyddion software. 

### 2.11. Cell Culture

SH-SY5Y cell line was grown in DMEM medium (Sigma, St. Louis, MO, USA) supplemented with 10% heat-inactivated fetal bovine serum (FBS), glutamine (2 mM), penicillin (100 U/mL) and streptomycin (100 mg/mL) in a humidified incubator with 5% CO_2_ in air at 37 °C.

### 2.12. Cytotoxicity Assays

Cell proliferation of SH-SY5Y inhibition was investigated according to sulforhodamine B (SRB) assay. Briefly, cells at 70–80% confluency were treated with trypsin (0.25% trypsin with 1 mM EDTA), washed, and resuspended in growth medium; 100 μL of cell suspension (10^5^ cells/mL) was added to each well of a 96-well plate. After overnight incubation, cells were treated with different concentrations of AlgOX (1–400 μg/mL), AlgOX-DA, and DA for 24 h. After treatment, cells were fixed in 10% trichloroacetic-acid (TCA) and stained with 0.4% (*w*/*v*) SRB dissolved in 1% acetic acid. SRB was removed and the plates were washed four times with 1% acetic acid to remove unbound dye before air-drying. Bound SRB was dissolved in 200 μL of 10 mM unbuffered Tris-based solution and plates were left on a plate shaker for at least 10 min. Absorbance was measured in a 96-well plate reader at 492 nm.

The absorbance ratio of treated to untreated cells allowed the calculation of the percentage of cell survival. The data presented are means ± standard deviation (S.D.) from eight replicates of five independent experiments. Viable cells were also counted by the trypan blue exclusion assay and light microscopy.

### 2.13. Statistics

For release studies and mucoadhesion tests, statistical analyses were carried out using Prism Version 4, GraphPad Software Inc. (San Diego, CA, USA). Data were expressed as either mean ± SD. Multiple comparisons were based on one-way analysis of variance (ANOVA) with the either Bonferroni’s or Tukey’s post hoc test, and differences were considered significant when *p* < 0.05. 

For biological studies, experimental points represented means ± SD of three to five independent experiments, performed in eight replicates, undergoing statistical evaluation using the ANOVA test. Where indicated, post hoc tests (Bonferroni/Dunn) were also performed.

## 3. Results

### 3.1. Synthesis of AlgOX and AlgOX-DA and Raman Spectroscopy Identification 

The synthesis of AlgOX was carried out by oxidizing NaAlg with NaIO_4_, and oxidation occurred over 24 h on the –OH groups at C2 and C3 of the of the M and G residues with main oxidation of the G residues [33]. Being a photosensitive reaction, the procedure was performed in a container with darkened walls [26], where, through the use of a 1:1 mixture (water:ethanol, *v*/*v*), the final yield achieved was 55% rather than 35%, as was the case when only water was adopted, according to the report of Balakrishnan et al. [34]. These authors showed that, in addition to an increase in oxidized product yield, when the reaction is carried out in a water:ethanol mixture, extensive polymer cleavage also occurred. All these effects were rationalized by invoking the formation of more reactive hydroxyethyl radicals during the oxidation reaction [34]. Concerning spectroscopic characterization, the Raman spectrum in Figure 2 shows the characteristic peaks of NaAlg [35,36] where symmetrical and asymmetrical stretching vibrations of –COO^−^ at 1420 and 1600 cm^−1^ occurred together with a C–O stretch at 1330 cm^−1^ and a C–O–C stretch at 1100 cm^−1^.

When Raman spectra of NaAlg and AlgOX were compared (Figure 2), in the case of AlgOX, the peaks corresponding to the ones of NaAlg were identified, together with a lowering of those at 1400 and 1050 cm^−1^, demonstrating the oxidation of the monomer M of the NaAlg backbone [34]; the peak at 1300 cm^−1^ indicated oxidation of the G monomer.

### 3.2. SEM-EDX Investigations

SEM microscopy is a useful method to analyze the morphology and formation of micro and nanostructures on a surface. In Figure 3a,b NaAlg and AlgOX show similar external morphologies; from Figure 3c it can be deduced that the surface of AlgOX-DA seems thinner than those of NaAlg and AlgOX. Moreover, on the surfaces of AlgOX and AlgOX-DA, no micro- or nanostructures were detected. For NaAlg, AlgOX, and AlgOX-DA (Figure 4a–c), the recorded EDX spectra were combined with elemental analyses of the sample composition. The analyses of NaAlg and AlgOX (Figure 4a,b) showed that both samples had the same ratio of O/C, equal to 2, leading to the conclusion that NaAlg oxidation to AlgOX occurred. As a matter of fact, the oxidation from alcohol function to aldehyde only involves the loss of H atoms, so the ratio stays the same. Elemental analysis calculations of AlgOX (Figure 4b) led to achieving the following values for oxygen, carbon, and nitrogen: 52 ± 2, 33 ± 3 and 0.03 ± 0.07. For AlgOX-DA (Figure 4c) they were 45 ± 4, 46 ± 6, and 3 ± 2. The increase in the carbon percentage and the presence of nitrogen atoms were ascribable to DA, whereas the decrease of the oxygen atom content was due to the water loss that occurred when the imine bond was formed. Therefore, these findings confirm the formation of a Schiff base between AlgOX and DA.

### 3.3. UV-Vis and FT-IR Analysis

UV-vis spectra were used to gain information about the covalent linkage of DA to the NaAlg backbone. The UV-vis spectrum of the AlgOX-DA imine conjugate, as well as those of AlgOX and pure DA, are shown in Figure 5. ForAlgOX and AlgOX-DA UV-vis spectra, the initial part (250 nm) was too sharp and such effects seemed to be ascribable to the presence of water solvent. As can be seen, a characteristic peak appeared at 280 nm, attributable to the absorbance of pure DA. In the case of the AlgOX-DA imine conjugate a shoulder was observed at 287 nm, while neither the absorption peak nor the shoulder were detected in the corresponding spectrum of AlgOX. The shoulder at 287 nm may have been due to the successful linkage between AlgOX and DA, the absence of any peak at 320 and 425 nm also meant the absence of quinone and polydopamine structures, respectively [37].

Concerning FT-IR spectra (Figure 6) for the DA spectrum, the strong and overlapped peaks between 1615 and 1584 cm^−1^ were ascribable to the bending of N–H of the DA primary amine group and the stretching vibrations of C–C belonging to the aromatic ring. Both spectra of AlgOX and AlgOX-DA showed a broad band at 3400 cm^−1^, ascribable to –OH groups of the polysaccharide vibrations and, in the case of FT-IR spectrum of AlgOX-DA, the O-H group of the catechol ring should also be included in that band, as previously demonstrated by Yue et al. [37]. Moreover, in both FT-IR spectra, instead of the typical stretching of C=O at 1700–1725 cm^−1^, the bands at 1622 cm^−1^ and at 1636 cm^−1^ were indicative of the symmetric stretching vibration for the carbonyl group for the aldehyde and the Schiff base groups, of AlgOX and AlgOX-Da, respectively. 

### 3.4. NMR Spectroscopy

The signals at 5.3–5.2, 5.1–4.5 and 4.3–3.7 ppm were assigned to protons of mannuronic acid and guluronic acid units (Figure 7a) [26]. The oxidation of Alg to Alg OX was confirmed by the signal at 5.4 ppm (Figure 7a), which corresponded to a hemiacetalic proton formed from aldehyde and the neighboring hydroxyl group, and by the absence of a signal at 9.5–10 ppm of a free aldehyde group, in agreement with Gomez [26].

The soluble macromolecules of AlgOX-DA were characterized in D_2_O solutions using ^1^H-NMR (see Gao [38]) in order to increase the solubility of AlgOX-DA in D_2_O, the NMR tube was preheated at 40 °C. 

The signals of Figure 7b, located at 6.77 (1H, d, *J* 7.8 Hz, *H-1* in Figure 7b), 6.71 (1H, d, *J* 1.8 Hz), 6.63 (1H, dd, *J* 1 7.8 Hz, *J* 2 1.8 Hz), 3.10 (2H, t, *J* 7.8 Hz), and 2.74 (2H, t, *J* 7.8 Hz) ppm correspond to DA protons. In the ^1^H-NMR spectra of AlgOX-DA (Figure 7b) additional peaks appeared. In ^1^H-NMR spectrum of AlgOX-DA (Figure 7b) additional peaks appeared, with respect to the ^1^H-NMR spectrum of AlgOX (Figure 7a), deriving from the protons in aromatic ring of the bonded DA (6.6~6.8 ppm) and from the imminic proton (HC=N 7.5 ppm).

Comparing the integral of HC=N (7.5 ppm) with that of the *H-1* unreacted DA signal located at 6.77 ppm, the percentage of incorporated DA in AlgOX-DA was 0.11%, which corresponded to 0.04 mmol of neurotransmitter. Working with 0.127 mmol of initial monomer-type units of AlgOX, the determination of the resulting degree of substitution (DS) was 31%, as determined by ^1^H-NMR spectroscopy. The amount of DA covalently bound to the AlgOX-DA imine conjugate was also determined by hydrolysis under strong acidic conditions, as described in Section 2.6, and it was found to be equal to 36 μg of DA/mg of AlgOX-DA imine conjugate.

### 3.5. Thermal Analysis via DSC

Prior to performing DSC analysis, melting point values were acquired for AlgOX and AlgOX-DA. Precisely, AlgOX started to decompose at temperatures higher than 200 °C and, at temperatures higher than 250 °C, decomposition also started for AlgOX-DA. Information on the solid state of AlgOX-DA was obtained by comparing its DSC thermogram with those of AlgOX and pure DA (Figure 8). The DSC thermogram of DA is characterized by a sharp endothermic peak at 250 °C, corresponding to the its melting point. The DSC curve of the AlgOX-DA imine conjugate also showed an endothermic peak at 130 °C, even though it was less intense, probably due to the loss of the physically bound water on the polysaccharide derivative. The thermogram of AlgOX showed an endothermic peak at 138 °C plus two more peaks at 127 °C and 162 °C. These last peaks, following literature suggestions, should be due to the loss of water associated with the hydrophilic groups of NaAlg [39]. In the case of the peak at 128 °C, such a temperature value was the end set of an endothermic peak, ascribable to the backbone of non-oxidized NaAlg [39].

### 3.6. XPS Studies

X-ray photoelectron spectroscopy analyses were performed to gain insight into the surface chemical composition of the novel AlgOX-DA conjugate. DA and AlgOX were also measured, being starting materials and useful for controls. XPS elemental percentages are reported in Table 1, while carbon and nitrogen high resolution spectral regions are depicted in Figure 9.

As expected, carbon, oxygen, and nitrogen were detected on the surface of pure DA, along with chlorine due to the hydrochloride form of dopamine. The surface of AlgOX showed residual traces of the NaIO_4_ used in the oxidation protocol of alginate. Nevertheless, the AlgOX-DA showed a good amount of surface nitrogen, demonstrating the successful preparation of the conjugate. This was further confirmed by detailed C1s and N1s investigations. Figure 9a shows a comparison of curve-fitted C1s of AlgOX and AlgOX-DA, with the attribution of the carbon chemical environments. A similar evaluation is presented in Figure 9b, where the curve-fitted N1s of AlgOX-DA is compared with the N1s of DA. In both cases, the spectra of the conjugate presented peak components, univocally assignable to the formation of the Schiff base; that is, the –C–N peak at 285.7 ± 0.1 eV in C1s spectrum, and the -N=C at 400.1 ± 0.1 eV in N1s region.

### 3.7. Mucoadhesion Studies

Among the different methods for in vitro evaluation of mucoadhesive effects, herein, the changes in transmittance values of mucin upon contact with AlgOX and AlgOX-DA were monitored in order to assess the rank order of the mucoadhesive materials [26]. As shown in Figure 10, from the turbidimetric measurements, the mucoadhesive properties of the tested materials could be classified in the following rank order: AlgOX-DA > AlgOX > HEC.

Although HEC was already employed as a reference semisynthetic polymer due to its good mucoadhesive features [40], from 30 min to 24 h, the transmittance values detected at the wavelength of 650 nm for Alg OX-DA were characterized by a statistically very significant difference (*p* < 0.01) vs. the control HEC (Figure 10). From turbidimetric measurements, the decrease in transmittance vs. HEC after 24 h of incubation resulted in being greater in the case of AlgOX-DA imine conjugate than for AlgOX. The corresponding differences were evaluated using the ANOVA approach and both Bonferroni’s and Tukey’s post hoc tests. It was found that the difference between AlgOX vs. HEC was not statistically significant (*p* > 0.05) while the difference of AlgOX vs. AlgOX-DA imine conjugate was significant (*p* < 0.01) as was that of AlgOX-DA imine conjugate vs. HEC, with *p* < 0.01.

### 3.8. In Vitro DA Release from AlgOX-DA in SNF and PBS

The behavior of AlgOX-DA to the chemical hydrolysis in the release the neurotransmitter was evaluated under two different pH media. In Figure 11a, the release profile of AlgOX-DA was determined in simulated nasal electrolyte solution without enzymes buffered at pH 6, where 13% of DA was delivered in 24 h, according to a sustained release trend. Precisely, the receiving medium changed color from clear to pale yellow in the frame time of 3 h to 24 h; perhaps in good agreement with progressive Schiff base hydrolysis. According to the literature, it has already been seen that, under the conditions used (pH 6.0 without enzymes), further oxidative products of the neurotransmitter DA are almost absent [41]. The diffusion from a kinetic point of view may be described by the Higuchi equation (i.e., C_T_ = K_H_ × t_1/2,_ where C_T_ is the amount of DA at time t) and the degradation reactions, which may be expressed by a first order reaction rate. The resulting global reaction rate can be simplified to the Higuchi equation only for very short time or for very slow degradation reactions. Following this approach, by plotting the percentages of cumulative DA release against the square root of the time, the Higuchi parameter (K_H_) can be obtained. When PBS was adopted as the receiving medium, then autoxidation of DA due to the higher pH value is of aid to explain the percentage reduction of DA delivered from 5 h (Figure 11c), reaching around 5% of neurotransmitter in the medium at 24 h. In the case of the AlgOX-DA imine conjugate in SNF, a KH value of 7.476 ± 1.828 (% of DA released × h^−1/2^) was calculated for the first 3 h of release. On the other hand, for the first 3 h of release in PBS (pH 7.4) the calculated K_H_ value was 11.90 ± 4.563 (% of DA released × h^−1/2^). From these results, it appears that, under acidic conditions (i.e., SNF), a sustained neurotransmitter release occurs while, in neutral/alkaline conditions (i.e., PBS), a faster release was observed.

### 3.9. FITC Labeling of AlgOX

The labeling procedure described for AlgOX was similar to that followed for some chitosan derivatives, such as fluorescent carboxymethylchitosan already preparedby us [28] and fluorescent glycol chitosan [31]. The resulting fluorescent AlgOX was obtained as an orange powder with a labeling efficiency of 2.8 μg FITC/mg FITC-AlgOX conjugate with a yield of the final fluorescent biomaterial equal to 27% ± 3.

### 3.10. PCS Analysis and SNOM Visualization

According to PCS methodology, the particle sizes of AlgOX and AlgOX-DA were determined (Table 2). Interestingly, the mean diameter of the imine conjugate AlgOX-DA was lower than that of AlgOX, although the polydispersity index, which is correlated to the colloid distribution, was similar and slightly high for both materials (Table 2). Furthermore, visualization of fluorescent AlgOX was also performed via SNOM microscopy. Figure 12b shows a fluorescence SNOM image with the corresponding shear-force topography (Figure 12a) of an FITC-AlgOX sample at a concentration of 1 mg/mL, taken as such. In the SNOM microphotographs, lighter areas are typical of a stronger emission. The shear-force image shows microstructures with a width of a few microns and a height of several tens of nm (Table 3). Similar topographic images were observed through all samples and may be considered representative of AlgOX structure [42]. The fluorescence spectroscopic SNOM images show emission distribution in areas that corresponds to the microstructures and the largest emission is always observed around the FITC-AlgOX microstructures. From these observations it is possible to state that there is almost a complete one-to-one correspondence between topographical microstructures and maxima in fluorescence emission, indicating that most of the fluorophore was stuck to AlgOX backbone, evidencing uniform fluorescence distribution throughout the sample. Additionally, in a few areas in Figure 12a,b, a fluorescence emission is shown, even though no topographical microstructures were observed. A similar fluorescence behavior was observed when FITC-AlgOX was processed at a concentration of 0.01 mg/mL (Figure 12c,d), even though it was very difficult to find appropriate areas where a microstructure gave raise to fluorescence signal, due, perhaps, to the high level of dilution of the fluorescent probe. A statistical analysis of FITC-AlgOX (1 mg/mL) characteristics according to AFM-SNOM observations is summarized in Table 3.

### 3.11. Cell Viability

In this study, we confirmed that DA reduces the viability of SH-SY5Y cells in a dose-dependent (1–50 μM) manner (Figure 13a), but such cytotoxic effect decreased when DA was linked to AlgOX in the imine conjugate AlgOX-DA (Figure 13b). Notably, AlgOX completely reversed the effect on cell vitality of 10 μg/mL DA, although at concentrations above 100 μg/mL a significative reduction in cell viability was observed (Figure 13a). Moreover, FITC-AlgOX was incubated with SH-SY5Y for 24 h, and at any tested concentration no effect on SH-SY5Y cell viability was observed.

## 4. Discussion

The aim of the present work was to synthesize and characterize a novel macromolecular AlgOX-DA imine conjugate and to evaluate its potential intranasal DA delivery by an AlgOX-based hydrogel formulation as an innovative PD treatment approach. It should be taken account that (i) intranasal administration may constitute a valuable non-invasive strategy to bypass the BBB hurdle; (ii) hydrogels have attracted considerable attention for intranasal administration; and (iii) the SH-SY5Y cell line has been used as a suitable experimental model for studying PD.

The conjugate AlgOX-DA herein proposed for nose-to-brain delivery was obtained after initial NaAlg oxidation according to [34]. Using Raman spectra, the oxidation of starting Alg occurred to produce AlgOX (Figure 1). Regarding the identification of AlgOX, as mentioned in Section 2.3, we assigned the structure of this polymer by comparison with the FT-IR spectrum reported in literature for an authentic sample (Figure 6). In particular, in this spectrum, the stretching absorption band of the aldehyde group was not detected, however, this is in agreement with literature reports [24]. It has been shown that aldehyde adsorption is not observed in dry samples of AlgOX-DA, while it can be detected in moist samples equilibrated at r.t. In this case, an equilibrium between aldehyde and hemiacetal groups occurs [24]. Concerning the structural assignment of the AlgOX-DA imine conjugate, once again, we did not observe the absorption band of the imine group expected in the range 1640–1690 cm^−1^ in the corresponding FT-IR spectrum. Although for AlgOX-DA the large band at 1636 cm^−1^ in the FT-IR spectrum can also be interpreted as the absorption of carboxylate groups of AlgOX overlapping with the Schiff base, the bands in the FT-IR spectrum help in the interpretation of the chemical structure of the conjugate AlgOX-DA. In this last spectrum, it is possible to detect a weak absorption band at 2915 cm^−1^ and 2917 cm^−1^ for AlgOX and AlgOX-DA, respectively, which may be attributed to the H–C absorption of the aldehyde group, which is present in the chemical formulas of both the substances. On the other hand, the UV-vis and the ^1^H-NMR spectra clearly support the structural assignment made, particularly the latter spectrum, where the resonance of the imine group was detected at 7.5 ppm. The presence of peaks that can be ascribed to the DA moiety are in the range of 6.6–6.8 ppm (aromatic portion of DA) and 2.74–3.10 ppm (aliphatic portion of DA). It should be pointed out that the chemical shift of the imine group we found resulted in a good agreement with that reported in the literature for other –CH=N containing conjugates. Moreover, elemental analysis combined to EDX also confirmed the composition of the conjugate AlgOX-DA and SEM microphotographs evidenced a similar external structure between pristine NaAlg and AlgOX-DA. In good agreement with such evidence, XPS data acquisition and spectra processing (Figure 9 and Table 1), following protocols that have been previously described [43], also led to the conclusion that the synthesis of the imine based conjugate AlgOX-DA was successfully carried out.

Furthermore, the DS of the final AlgOX-DA was achieved following either NMR calculations on the corresponding monodimensional spectrum or performing acid-induced hydrolysis followed by HPLC quantification of the delivered neurotransmitter. Of note, the DS of AlgOX-DA determined after chemical hydrolysis allowed quantitative evaluations and, therefore, DS was employed for in vitro release studies and AlgOX-DA exposure to the cell line. For instance, for AlgOX-DA, the DS for DA we found was almost double that previously studied for a Alg hydrogel containing DA, namely 15.86 ± 2.04 μg DA/mg DA-hydrogel already studied for drug delivery applications [38]. 

In order to explore the potential of AlgOX and AlgOX-DA for nasal adaptability, in vitro investigations focusing on mucoadhesion and release were carried out (Figure 10). Turbidimetric measurements were carried out in the presence of a positive control of hydroxyethyl cellulose (HEC) because Carbopol 940 precipitated in SNF medium, as previously shown [44]. Of note, during mucoadhesion studies referring to AlgOX and AlgOX-DA, no color change at any time of incubation was evidenced for both biomaterials, different from our previous investigations on another polymeric prodrug of DA; namely, a DA-ester-derivative of *N*,*O*-carboxymethyl chitosan, where at the last time point showed a black color in the SNF medium. Although the polymeric backbone is different (i.e., *N*,*O*-carboxymethyl chitosan vs. AlgOX), our hypothesis is that for both conjugates, during exposure to SNF, an autoxidation process of the neurotransmitter took place [28] once DA was released after hydrolytic cleavage in SNF. It might be faster for ester derivatives rather than for imine-based AlgOX-DA.

Interestingly, in vitro mucoadhesive studies in SNF showed that the conjugate AlgOX-DA was more mucoadhesive than unmodified AlgOX, determining a reduction in percentage of transmittance of up to 78% within 24 h of mucin/AlgOX-DA contact (Figure 10). Such behavior can be explained in terms of the catechol moiety of the neurotransmitter that, in the AlgOX-DA backbone, improves the mucoadhesive effects. Chitosan-catechol-containing conjugates have already been described for the enhancement of catechol moiety in terms of mucoadhesion [45,46] and it seems that the same role could be exerted here, irrespective of the fact that the backbone is Alg-based. Such findings also seem in good agreement with Nur et al.’s investigations [47], evidencing that the prediction of mucoadhesive effects based on rheology measurements indicated that, among mucin–polymer mixtures, chitosan and polyvinyl pyrrolidone (PVP) exhibited a lower strength of mucoadhesion compared to Alg and its derivatives. On the other hand, for AlgOX-DA, such a high mucoadhesive effect is combined with a sustained release capability, as shown in Figure 11a, where the hydrolysis of the Schiff base leads to a delivery of 13% of the neurotransmitter in 24 h in SNF, in agreement with the sensitivity of the Schiff base to undergo the hydrolysis process under acidic conditions. Nevertheless, free DA may slowly be further converted into degradation products and, therefore, we studied these consecutive reactions from a kinetic point of view according to Juriga et al.’s approach [48] for polymer–DA conjugates. In particular, for poorly soluble conjugates, such as the AlgOX-DA imine conjugate, DA release can be expressed by the diffusion of DA from the solid surface of the conjugate combined with the degradation reactions of DA in solution [48]. In the case of the AlgOX-DA imine conjugate, a K_H_ value of 7.476 ± 1.828 (% of DA released × h^−1/2^) for the first 3 h of release was calculated, indicative of a sustained neurotransmitter release.

Another critical feature for a DA-loaded formulation intended for nose-to-brain delivery for Parkinsonian patients is that the neurotransmitter is released in the dosage form in sustained manner so that a reduced stimulation of dopaminergic neurons occurs [28]. As shown in Section 3.8, in SNF, the neurotransmitter is released from the AlgOX-DA imine conjugate in a sustained fashion (as proved by the observed K_H_ value) within the first 3 h, even though it is about 13% of the total amount. This low amount of neurotransmitter released in a sustained manner, may even be advantageous for the decreased neuronal toxicity [28]. Moreover, it should be also considered that (i) DA, as neurotransmitter, is a potent substance able to produce its biological effects, even if present in small amounts; (ii) the conditions we used for these in vitro release studies cannot be considered fully biomimetic and, hence, definitive information on DA release from AlgOX-DA imine conjugate can only be gained by performing in vivo studies. On the other hand, when PBS was used as a receiving medium for in vitro release tests (Figure 11c), its physiological pH value is of aid to understand that the clear decrease in neurotransmitter release observed at longer times should likely be ascribed to the most likely autoxidation reaction of the neurotransmitter in neutral/alkaline conditions. Preliminary tests of AlgOX-DA stability in fetal bovine serum/PBS (1:1, *v*/*v*) were started by us in order to predict the behavior of the conjugate once absorbed by blood vessels (unpublished data). Three percent of DA was found to be delivered within 3 h and, once data are replicated, such findings could mean that an almost intact polymeric prodrug AlgOX-DA could by distributed from capillary vessels to the *Substantia Nigra* in the brain, providing a site-specific delivery of the neurotransmitter. 

Overall, delivery systems allowing sustained release of a neurotransmitter, as seen for AlgOX-DA, may be useful to avoid multiple administrations of L-Dopa, which is currently the gold standard for PD treatment. In addition, if a sustained release of DA can be achieved in vivo by replacing L-Dopa administrations [49], a reduction in the toxic effects of L-Dopa can be expected, due to the fact that they typically emerge at later stages of PD. Overall, in view of nose-to-brain delivery, mean particle size of AlgOX-DA (Table 2) seems promising for bypassing the BBB because, as with colloidal carriers aiming at such non-invasive absorption approach, it has already been discussed that 200 nm is regarded as a suitable value for mean diameter [50,51].

From a biological viewpoint, it is well known that DA induces a loss of cell viability associated with mitochondrial dysfunction in SH-SY5Y cells [52], thus the strategy of bioconjugation was selected by us in an attempt to overcome such limitations. Therefore, in view of cell/conjugate interaction studies, fluorescent dye FITC was condensed on AlgOX and cell viability studies with SH-SY5Y (Figure 13b) evidenced that FITC-AlgOX can be considered a safe probe material if incubated for 24 h up to a concentration of 450 g/mL, whereas unlabeled AlgOX showed cytotoxicity at 400 g/mL (Figure 13a). 

Concerning the microscopic evaluation of FITC-AlgOX, the SNOM apparatus employed worked with a resolution that depended on the aperture dimension of the optical fiber rather than the wavelength of the light, as is the case in conventional optical fluorescence spectroscopy. Therefore, only a small volume of the sample was optically illuminated, in order to minimize fluorophore bleaching and, consequently, the biological damage normally occurring in optical fluorescence microscopy was reduced. Herein, the use of a localized optical microscope, such as SNOM, allowed a deep investigation in the features of fluorescently labeled AlgOX samples, focusing on targeted areas of the sample. The SNOM characterization of the FITC-AlgOX fluorescent effect (Figure 12) evidenced a homogeneous distribution within the conjugate, either at 1 mg/mL or 0.01 mg/mL concentrations, which were safe concentrations of FITC-AlgOX upon incubation with SH-SY5Y cell lines.

Generally, biological samples are characterized by their intrinsic variability, such as differences in shape or functionality in the same cell culture population. The SNOM microscope is well known to provide high-resolution optical images, including nanometer-scale topographic information [53,54,55], and is of aid in the examination of biological interactions, allowing a fast and direct identification and analysis of specific features. Finally, SNOM images and characterization of FITC-AlgOX alone obtained herein will be available for further studies, aiming at monitoring topographical and optical characteristics upon cell incubation.

## 5. Conclusions

In this work, investigations focusing on the conjugation of DA via an imine bond to AlgOX were performed. Multiple spectroscopic techniques, namely, ^1^H NMR, UV-vis, FT-IR and X-ray photoelectron spectroscopic methods, confirmed the presence of the Schiff base for the AlgOX-Da conjugate and the slow release of DA was revealed after incubation in SNF and in PBS. The conjugate showed mucoadhesive properties in vitro that were better than those of pristine polymer Alg, as well as a sustained release of DA in SNF at about 13%; an amount that may be advantageous for a decrease in neuronal toxicity and sufficient to perform its biological effects given the high potency of the neurotransmitter. Furthermore, the cytocompatibility of the imine conjugate was assessed using the SRB assay against human neuroblastoma SH-SY5Y cells in a wide range of concentrations. Our work demonstrates that the AlgOX-DA imine conjugate is a promising delivery system for non-invasive nasal administration, as an innovative approach for PD treatment, using a hydrogel dosage form or polymeric nanoparticles [43].

## Data Availability

The data presented in this study are available on request from the corresponding authors. The data are not publicly available due to ethic reasons.

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
