# Peer review of "Oxidized Alginate Dopamine Conjugate: In Vitro Characterization for Nose-to-Brain Delivery Application"

_materials, 2021, doi:10.3390/ma14133495_

Round 1

Reviewer 1 Report

The manuscript entitled “Oxidized Alginate Dopamine conjugate: in vitro characterization for Nose-to-Brain delivery application” submitted by Cataldo, Bellucci and collaborators describes the synthesis and characterization of oxidized alginate dopamine imine conjugate and the evaluation of its mucoadhesive properties, DA release and cytotoxicity in humans.

The language used in the manuscript is correct. However, I recommend to proof-read again the text in detail as many typos can be found. For example, to mention some:

Page 3 line 134: TEA (55 l)

Page 5 line 181: (  )

Page 7 line 294: of the of the

Figure 5 and 6 captions: blue should read instead of blu

Page 13 line 381: 5.15-⌐5.4 ppm

Page 16 line 435: Chlorhydrate should read instead of chloridrate

Page 24 line 661: in in

The figures included in the manuscript are of poor quality. The images show very different quality and formatting among them. I suggest the authors to improve the quality and homogenenize the format of all of them. For example, the quality of Scheme 1 is bad. Figure 7 shows 1H NMR spectra of different compounds. It occupies 2 pages and the comparison between spectra can be done in one single axis. The same occurs for figure 2.

The interpretation and discussion of the experimental results lacks of accurate explanations. Below, I suggest a list of changes and experiments that could help the authors improving the contents and comprehension of the manuscript for next submissions.

  1. The final yields described different yields when using water or water:ethanol mixture. Please add some comments in the rationale for this effect.
  2. Raman spectra of NaAlg and AlgOx shown in figure 2 are very similar. Which are the main differences to highlight? Did the authors performed the Raman of the DA modified? Is there any characteristic band that demonstrates the functionalization of AlgOx?
  3. I suggest showing all 1H NMR spectra with the same spectral width. Moreover, I recommend the authors to add the proton assignment of the most relevant protons and describe the changes in the spectra before and after DA linkage. This part is very confusing in the text.
  4. Figure 3 a and c are overlapping.
  5. The EDX spectrum c in figure 4 shows peaks for Pd and F. Could the author comment on that?
  6. Also for the EDX spectra, the authors comment on the Oxigen/Carbon ratio to demonstrate the functionalization of the ArgOx. However, the peak of Carbon can be masked by the presence of other elements in this area.
  7. In which solvents are the UV-Vis of AlgOX and AlgOX-DA performed? Please ensure that the solvent is not absorbing in this range. The initial part of the UV-Vis spectra (250 nm) is too sharp to be from the absorbance of the compound.
  8. FT-IR spectra of AlgOX and AlgOX-DA do not show a lot of differences. From my point of view, these results are not conclusive to demonstrate the immobilization of the DA.
  9. I wonder if mucoadhesive effects in SNF should vary along time. Figure 10 does not show any trend along time.
  10. Figure 11 shows DA cumulative release vs time. Please describe how do you calculate the cumulative release in this specific case.
  11. In Figure 13, what does a, a’, b, b’ and c refers to?

For all the stated above I can consider the re-submission of the manuscript after addressing the issues raised above.

Reviewer 2 Report

The authors demonstrate conjugation of oxidized alginate hydrogel to dopamine via imine crosslinking for potential application of the material for intranasal delivery to the brain tissues. The chemistry is based on a previous work on conjugating alginate with antitumor drugs. This approach is interesting and has potential for further research and development in the field of neurotherapeutics which is hampered by a critical issue of overcoming the blood-brain-barrier. The characterization and discussion is sufficient. However, the authors should address some of the below issues.

  • Line 181: check the empty brackets in ..parts per million ( )
  • The conjugation seems working as per several other molecular characterizations, however, the downside is that the authors do not directly measure the morphology of the synthesized nanoparticles. Characterization of these properties by TEM could be more accurate.
    1. Are these uniformly sized conjugated particles?
    2. Is the drug supposed to be embedded inside a nano-alginate shell?
    3. In that case Would making the shell with different alginate grades affect the solubility and muco-release properties ?
  • Past the mucus layers, the nanoparticles are supposed to be absorbed into the blood vessels. In this case some more characterization of the particles in blood / simulated colloidal fluids would be necessary
  • Have the authors characterized the solubility of the conjugates?
  • The authors need to discuss more about the possible ideal formulations in terms of intranasal delivery applications, is the final form of the conjugate supposed to be a lyophilized powder or a spray or dispersion? If its delivered only as gel dispersion, would it require the gel to degrade to have the particles released into the mucosal layers?
  • The resolution of EDX spectra is very low. The figures need more clarity , perhaps some arrow indications/annotations can be added.
  • To overcome size-based limitations, the proposed overall size of the conjugates (after measuring accurately) needs to be compared against sizes of other drugs/formulation meant for intranasal delivery and discussed.

Author Response

Reviewer 2

  • Line 181: check the empty brackets in ..parts per million ( )

We apologize for the inconvenient occurred. The Reviewer’s request was matched in the revised version.

  • The conjugation seems working as per several other molecular characterizations, however, the downside is that the authors do not directly measure the morphology of the synthesized nanoparticles. Characterization of these properties by TEM could be more accurate

We thank the Reviewer for this remark and TEM visualization of pure biomaterials AlgOX and AlgOX-DA will be performed in a step forward work. 

  • Are these uniformly sized conjugated particles?

We thank the Reviewer for his/her suggestion and, in the revised version new studies based on Photoelectron spectroscopy analysis were performed. Details are reported in Sections 2.10, 3.10 and in the current Table 2. As stated at Lines 527-531 from the evaluation of polydispersity index both AlgOX and AlgOX-Da were endowed of slightly high particle distribution.

  • Is the drug supposed to be embedded inside a nano-alginate shell?
  • In that case Would making the shell with different alginate grades affect the solubility and muco-release properties ?

Investigations are still ongoing for nano particulate delivery systems based on AlgOX-DA. Mucoadhesive effects; solubility determinations and in vitro release studies are planned as future perspectives of this work in order to compare the properties of the biomaterial AlgOX-DA in itself with the corresponding ones of AlgOX-DA based nanoparticles.

  • Past the mucus layers, the nanoparticles are supposed to be absorbed into the blood vessels. In this case some more characterization of the particles in blood / simulated colloidal fluids would be necessary.

For AlgOX-DA our hypothesis is that, after mucoadhesive interaction with nasal district, via olfactory and trigeminal neural connections it could be addressed to the brain compartment. On the other hand, we have appreciated the Reviewer’s observations because the capillary absorption of the conjugate AlgOX-DA cannot be ruled out “a priori”. Hence, in the revised version, in Discussion Section (Lines 676-681) we have shortly mentioned some preliminary results obtained after incubation of AlgOX-DA in the mixture fetal bovine serum/PBS (1:1, v/v). The low percentage of DA delivered in such medium seems promising for maintaining the entire conjugate in the blood prior to reach the Substantia Nigra but further analysis is necessary to draw definitive conclusions.

I

  • Have the authors characterized the solubility of the conjugates?

In this work, a systematic solubility study concerning AlgOX and AlgOX-DA missed because we aimed at an extensive spectroscopic characterization of the novel materials AlgOX and AlgOX-DA in view of their pharmaceutical application.

Nevertheless, we have found that AlgOX is fully soluble in double distilled water at the concentration of 20 mg/mL. On the other hand, AlgOX-DA is fully soluble in phosphate buffer saline (pH = 7.4) at the concentration of 20 mg/mL, whereas its solubility in double distilled water is 0.25 mg/ml. Although the aqueous solubility of AlgOX-DA could appear a low value, as reported in Section 3.9, the soluble amounts of AlgOX-DA undergo to the cleavage of the Schiff in SNF, ensuring DA delivery in vitro, also in the absence of enzymes.

  • The authors need to discuss more about the possible ideal formulations in terms of intranasal delivery applications, is the final form of the conjugate supposed to be a lyophilized powder or a spray or dispersion? If its delivered only as gel dispersion, would it require the gel to degrade to have the particles released into the mucosal layers?
  • We thank the Reviewer for this comment and, indeed, being research on the conjugate AlgOX-DA going on as a development of the current manuscript, further information concerning the device/formulation for administration will be provided later on. Due to the fact that Alginate- based gels can be easily obtained in the presence of calcium chloride solutions, it could be expected that AlgOX-DA conjugate can form gels upon the same conditions because carboxylic groups on the backbone of AlgOX are still available for calcium ion coordination. Once release studies on the resulting gels are carried out, then the exact mechanism of delivery will be derived.

  • The resolution of EDX spectra is very low. The figures need more clarity, perhaps some arrow indications/annotations can be added.

According to the Reviewer’s suggestion, we improved the EDX spectra resolution at the maximum allowed by the software. Moreover, in the revised version for each EDX spectrum an inset was included, in order to distinguish the elements falling at low energy values.

  • To overcome size-based limitations, the proposed overall size of the conjugates (after measuring accurately) needs to be compared against sizes of other drugs/formulation meant for intranasal delivery and discussed
  • We agree with the point raised by the Reviewer and in the revised version (Lines 688-691) we state that particle size of AlgOX-DA matches the requirements for nasal absorption in order to overcome the Blood Brain Barrier. At this regard, to corroborate our statements, two more references were introduced in the revised version (i.e., 50 and 51).

Round 2

Reviewer 2 Report

The authors have made the recommended changes and addressed the issues in sufficient detail